# Comparison of Tepotinib, Paclitaxel, or Ramucirumab Efficacy According to the Copy Number or Phosphorylation Status of the *MET* Gene: Doublet Treatment versus Single Agent Treatment

**DOI:** 10.3390/ijms25031769

**Published:** 2024-02-01

**Authors:** Sung-Hwa Sohn, Hee Jung Sul, Bum Jun Kim, Dae Young Zang

**Affiliations:** 1Hallym Translational Research Institute, Hallym University Sacred Heart Hospital, Anyang-si 14066, Republic of Korea; iisupy@korea.ac.kr (S.-H.S.); glwjd82@naver.com (H.J.S.); 2Department of Internal Medicine, Hallym University Sacred Heart Hospital, Hallym University Medical Center, Hallym University College of Medicine, Anyang-si 14068, Republic of Korea; getwisdom1025@gmail.com

**Keywords:** tepotinib, paclitaxel, ramucirumab, copy number variation, phosphorylated MET

## Abstract

Although conventional combination chemotherapies for advanced gastric cancer (GC) increase survival, such therapies are associated with major adverse effects; more effective and less toxic treatments are required. Combinations of different anti-cancer drugs, for example, paclitaxel plus ramucirumab, have recently been used as second-line treatments for advanced GC. This study evaluated how copy number variations of the *MET* gene, *MET* mutations, and MET gene and protein expression levels in human GC cells modulate the susceptibility of such cells to single-agent (tepotinib, ramucirumab, or paclitaxel) and doublet (tepotinib-plus-paclitaxel or ramucirumab-plus-paclitaxel treatment regimens. Compared with ramucirumab-plus-paclitaxel, tepotinib-plus-paclitaxel better inhibited the growth of GC cells with *MET* exon 14 skipping mutations and those lacking *MET* amplification but containing phosphorylated MET; such inhibition was dose-dependent and associated with cell death. Tepotinib-plus-paclitaxel and ramucirumab-plus-paclitaxel similarly inhibited the growth of GC cells lacking *MET* amplification or MET phosphorylation, again in a dose-dependent manner, but without induction of cell death. However, tepotinib alone or tepotinib-plus-ramucirumab was more effective against c-MET-positive GC cells (>30 copy number variations) than was ramucirumab or paclitaxel alone or ramucirumab-plus-paclitaxel. These in vitro findings suggest that compared with ramucirumab-plus-paclitaxel, tepotinib-plus-paclitaxel better inhibits the growth of c-MET-positive GC cells, cells lacking *MET* amplification but containing phosphorylated MET, and cells containing *MET* mutations. Clinical studies are required to confirm the therapeutic effects of these regimens.

## 1. Introduction

In 2020, 1,089,103 new cases of gastric cancer (GC) were diagnosed worldwide [1]. GC is the fourth leading cause of cancer death (7.7% of all such deaths). The current treatment guideline for locally advanced unresectable, metastatic gastric, or gastroesophageal junction adenocarcinoma is palliative chemotherapy, usually employing a platinum-plus fluoropyrimidine regimen [2,3,4]. Ramucirumab monotherapy or in combination with paclitaxel is a second-line treatment option for patients with advanced gastric or gastroesophageal junction adenocarcinoma (with or without hepatocyte growth factor receptor [MET]-positive tumors) [5,6]. It is accepted that conventional chemotherapy improves survival and the quality of life of patients with metastatic/recurrent GC. However, the 5-year survival rate remains low, and the only effective targeting therapeutics are trastuzumab and ramucirumab [7,8].

The c-MET receptor and its ligand, hepatocyte growth factor (HGF), are involved in carcinogenesis and metastatic tumor progression [9]. Point mutations activating and amplifying c-*MET* expression, and c-*MET/HGF* co-expression, have been observed in many human cancers, including GC [10,11,12]; high-level *HGF* expression and c-*MET* overexpression are often associated with poor clinical outcomes, thus more aggressive disease, increased tumor metastasis, and reduced survival [13,14,15]. The Cancer Genome Atlas (TCGA) project reported high-level changes in c-MET pathway activities among GC patients [16]. Of 157 intestinal gastric tumors evaluated by TCGA researchers, 3.2% harbored *MET* amplifications. Of 61 diffuse gastric tumors evaluated, 4.6% exhibited *MET* amplifications.

Tepotinib is a potent, highly selective, type Ib c-MET inhibitor with a favorable human pharmacokinetic profile, and once-daily dosing is possible [17]. Tepotinib inhibits cancer cell growth and induces regression of susceptible HGF-dependent and -independent tumors. Tepotinib efficacy and safety were assessed in a phase 1 single-agent study enrolling patients with solid tumors [18]. Previously, we explored (both in vitro and in vivo) whether tepotinib suppressed the activation of downstream signaling pathways, GC proliferation, apoptosis, MET, and tumor progression. Tepotinib dose-dependently inhibited the growth of c-*MET*-amplified SNU620, MKN45, and Hs746T GC cell lines and induced apoptosis [19,20]. 

Here, we explored whether tepotinib-plus-paclitaxel exhibited synergistic effects, and whether such treatment was more effective than ramucirumab monotherapy or ramucirumab-plus-paclitaxel in killing GC cells with *MET* exon 14 skipping mutations or *MET* amplification. We characterized the carcinogenetic potential of human GC cells and the effects of copy number variations (CNVs) and mutations in drug response-related genes in five human GC cell lines, as well as the susceptibility of such cells to single-agent (tepotinib, ramucirumab, or paclitaxel) or doublet (tepotinib-plus-paclitaxel or ramucirumab-plus-paclitaxel) treatments.

## 2. Results

### 2.1. The Cell Lines

Five GC cell lines (Hs746T, MKN45, SNU620, AGS, and SNU638) were analyzed in terms of the expression levels of 286 relevant genes by the Theragen Bio Institute (Seongnam, Republic of Korea) [20]. The 1000 Genomes data were used to filter out common germline variants that are not pathogenic. Genes exhibiting low and modified SnpEff impact figures were removed. We selected genes that met the GATK Mutect2 Variant Calling filter PASS criteria, i.e., those with high SnEeff impact values (Figure 1). The 286 genetic aberrations are shown in Figure 1.

*MET* mRNA was significantly overexpressed in the MKN45 and SNU620 cell lines compared to Hs746T, AGS, and SNU638 cell lines (Figure 2a). However, in previous studies, it was confirmed that Hs746T, MKN45, and SNU620 cells had more than 30 copies [20]. These Hs746T, MKN45, and SNU620 cell lines were *MET*-amplified cells, whereas the AGS and SNU638 cells were non-amplified cell lines (0 copies or not detected) (Figure 2b). We also observed Hs746T, MKN45, and SNU620 cells that overexpressed p-MET and MET protein, whereas the AGS and SNU638 cells had no or low-expressed p-MET and MET protein expression (Figure 2c).

### 2.2. The Effects of Ramucirumab, Paclitaxel, and Tepotinib on the Viability of Cells According to MET Expression

We explored the effects of ramucirumab, paclitaxel, and tepotinib on the growth inhibition of Hs746T, MKN45, SNU620, AGS, and SNU638 cells (Figure 3). Paclitaxel and tepotinib decreased cell viability in a dose-dependent manner, but ramucirumab had no effect. The paclitaxel-and-tepotinib combination optimally inhibited GC cell growth.

Non-linear regression analyses yielded the ramucirumab, paclitaxel, and tepotinib IC_50_ values (Table 1). 

We next evaluated the dose-dependent inhibitory effects of paclitaxel, 10 nM ramucirumab-plus-paclitaxel, 10 nM tepotinib-plus-paclitaxel, and 10 nM ramucirumab-plus-10 nM tepotinib-plus-paclitaxel on Hs746T, MKN45, SNU620, AGS, and SNU638 cells (Figure 4). Compared with ramucirumab-plus-paclitaxel, tepotinib-plus-paclitaxel better inhibited the growth of Hs746T cells with *MET* exon 14 skipping mutations and SNU638 cells lacking *MET* amplification but with phosphorylated MET in a dose-dependent manner. Tepotinib-plus-paclitaxel and ramucirumab-plus-paclitaxel similarly inhibited the growth of such cells. However, tepotinib alone, tepotinib-plus-paclitaxel, and tepotinib-plus-paclitaxel-plus-ramucirumab were more effective against c-MET-positive GC cells (Hs746T, SNU620, and MKN45; copy number variation [CNV] > 30) than were ramucirumab or paclitaxel alone or ramucirumab-plus-paclitaxel (Figure 3 and Figure 4). However, when tepotinib-plus-paclitaxel and tepotinib-plus-paclitaxel-plus ramucirumab were compared, there was no difference.

### 2.3. Effects of Ramucirumab, Paclitaxel, and Tepotinib on the Death of GC Cells According to MET Expression

We next investigated ramucirumab-, paclitaxel-, and tepotinib-induced death of Hs746T, MKN45, SNU620, AGS, and SNU638 cells by FACS-analysis of cell apoptosis and necrosis (Figure 5 and Appendix A). Compared with ramucirumab-plus-paclitaxel, tepotinib-plus-paclitaxel killed more GC cells with *MET* exon 14 skipping mutations or *MET* amplification (CNV > 30). However, tepotinib alone or tepotinib-plus-ramucirumab was more effective against c-MET-positive GCs (Hs746T, MKN45, and SNU620 cells) than was ramucirumab or paclitaxel alone or ramucirumab-plus-paclitaxel. In addition, SNU638 has a *MET* mutation and showed slight apoptosis induction when treated with tepotinib (Figure 1 and Figure 5).

### 2.4. The Effects of Ramucirumab, Paclitaxel, and Tepotinib on the Migration of GC Lines According to MET Expression

We next investigated the effects of ramucirumab, paclitaxel, and tepotinib on the migration of Hs746T, MKN45, AGS, and SNU638 cells (Figure 6). Compared with ramucirumab-plus-paclitaxel, tepotinib-plus-paclitaxel better inhibited migration. However, tepotinib alone and tepotinib-plus-paclitaxel were more effective against c-MET-positive GC (Hs746T and MKN45) cells than were ramucirumab or paclitaxel alone or ramucirumab-plus-paclitaxel.

### 2.5. Effects of Ramucirumab, Paclitaxel, and Tepotinib on Protein Levels in GC Cell Lines According to MET Expression

We measured the expression levels of carcinogenesis-associated proteins when determining the effects of ramucirumab, paclitaxel, or tepotinib on GC cells (Figure 7). Compared with ramucirumab-plus-paclitaxel, tepotinib-plus-paclitaxel better inhibited the expression of these proteins in GC cells with *MET* exon 14 skipping mutations or *MET* amplification. However, tepotinib-plus-paclitaxel was more effective against c-MET-positive GC cells than tepotinib alone.

## 3. Discussion

The molecular profiles of GC cells are important in terms of therapy and may usefully identify new therapeutic targets and clinically relevant biomarkers [21]. Of the four GC subtypes recognized by TCGA, tumors exhibiting chromosomal instability are associated with changes in the CNVs of *EGFR, EGFR2* (*HER2*), *FGFR2*, and *MET* [22,23]. Treatment guidelines are available for tumors expressing *HER2*; thus, we did not consider such tumors here. We sought mutations and CNV changes in/of 286 genes that affect carcinogenesis and drug responses (including the above three genes) in five GC cell lines (Hs746T, SNU620, AGS, SNU638, and MKN45) (Figure 1 and Figure 2). The Hs746T, SNU620, AGS, SNU638, and MKN45 cell lines evidenced only *EGFR* and *MET* mutations, and the *MET* CNVs were ≥ 5. MET activation promotes EGFR activity [24]; MET inhibition downregulates other pathways that promote carcinogenesis; treatment guidelines are required.

Various drugs targeting the c*MET* pathway have been used to control GC; no clinical benefit is yet apparent. Monoclonal antibodies targeting the c*MET* pathway (rilotumumab [25], onartuzumab [26], and emibetuzumab [27]) have been evaluated in randomized phase II or III trials enrolling patients with advanced GC, but no clinical benefit was apparent in any study. Patients were selected if they were immunohistochemically positive for MET; however, the positivity thresholds used are controversial. MET immunohistochemical data would not have reflected *MET* CNVs, but better results would have been obtained if *MET* CNVs had been reflected. Tyrosine kinase inhibitors targeting cMET have been evaluated in phase 2 clinical trials of GC patients [28,29,30,31,32]; the response rates ranged up to 33% [29], but the patient numbers were small. No study has yet identified a biomarker that reliably predicts the effects of treatment. Tepotinib has not been clinically evaluated in GC patients but has been trialed in lung cancer patients. In the VISION trial enrolling 152 non-small cell lung cancer patients, tepotinib afforded an overall response rate of 46% in patients with *MET* exon-14 skipping mutations [33].

Here, we compared the susceptibility of GC cells to single-agent (tepotinib, ramucirumab, or paclitaxel) and doublet (tepotinib-plus-paclitaxel or ramucirumab-plus-paclitaxel) drug regimens. As expected, the MET inhibitor tepotinib alone significantly reduced the viabilities of MET-positive MKN45 and SNU620 cells (Figure 3). However, tepotinib alone minimally reduced the viability of MET-positive Hs746T (*MET* exon 14 skipping mutations) cells. A combination of paclitaxel with either ramucirumab or tepotinib exhibited strong synergistic suppression of all GC cells (Figure 4). In particular, paclitaxel-plus-tepotinib reduced cell viability more effectively than paclitaxel-plus-ramucirumab. Paclitaxel-plus-tepotinib triggered greater apoptosis of MET-positive cells than did paclitaxel-plus-ramucirumab (Figure 5). Tepotinib-plus-paclitaxel better inhibited GC cell migration than did ramucirumab-plus-paclitaxel (Figure 6). Tepotinib alone, and tepotinib-plus-paclitaxel, strongly inhibited phospho-MET and PD-L1 expression in GC cells (Figure 7). Ramucirumab inhibits activation of the angiogenesis-related VEGFR2 protein [34]. We found that tepotinib-plus-paclitaxel better suppressed VEGFR2 than did ramucirumab-plus-paclitaxel. Tepotinib-plus-paclitaxel strongly suppressed the expression of ECAD, c-MYC, and phospho-AKT in MET-positive cells. A tepotinib-plus-paclitaxel regimen should become the standard second-line treatment for GCs exhibiting METex14SM mutations and GCs lacking *MET* amplification but evidencing MET phosphorylation.

## 4. Materials and Methods

### 4.1. Reagents

Ramucirumab, paclitaxel, and tepotinib were obtained from Selleck Chemicals (Houston, TX, USA). The annexin V-APC/propidium iodide (PI) apoptosis detection kit was obtained from Thermo Fisher Scientific (Waltham, MA, USA).

### 4.2. Cell Lines and Cell Culture

The cell lines, AGS, Hs746T, MKN45, SNU620, and SNU638, were obtained from the Korean Cell Line Bank (Seoul, Republic of Korea). The AGS, MKN45, SNU620, and SNU638 cells were cultured in RPMI 1640 medium (Thermo Fisher Scientific), and the Hs746T cells were grown in Dulbecco’s Modified Eagle Medium (Thermo Fisher Scientific). Both media were supplemented with 10% (*v*/*v*) fetal bovine serum and 1% (*w*/*v*) penicillin/streptomycin. All the cells were cultured using standard procedures.

### 4.3. Target Sequencing and Analysis

The target sequencing method has been described in detail [20]. The data from 1000 Genomes can be used to filter out common germline variants that are not pathogenic. During variant calling, sites of interest were overlapped with the SNPs from the 1000 Genomes Project Phase 3 dataset; results with a base quality ≥30 and related hits were filtered out.

### 4.4. Real-Time RT-PCR Analysis

A quantitative real-time RT-PCR technique was used to analyze the mRNA expression of *MET* and *GAPDH* in the five GC cells, as previously reported [20]. The transcript levels of GAPDH were used for MET normalization.

### 4.5. MTS Cell Proliferation Assay

To assess the effects of ramucirumab, paclitaxel, and tepotinib on cell proliferation, the MTS assay was performed using the CellTiter 96 Aqueous One Solution Cell Proliferation Assay kit (Promega, Madison, WI, USA). Briefly, AGS, Hs746T, MKN45, SNU620, and SNU638 cells were seeded into 96-well plates at ~50% confluence and incubated for 24 h. The cells were then treated with the drugs at 10, 1, 0.1, 0.01, 0.001, 0.0001, 0.00001, or 0.000001 µM for 48 h. Cell viability was assessed using the MTS assay. The IC50 values were calculated via nonlinear regression analysis employing Prism 5.0 software (GraphPad Software, San Diego, CA, USA).

### 4.6. Flow Cytometry

AGS, Hs746T, MKN45, SNU620, and SNU638 cells were seeded into six-well plates at 5 × 10^4^/mL and treated with various concentrations of ramucirumab (10 nM) paclitaxel (20 nM), or tepotinib (10 nM) alone or in combination. The concentration ranges were selected by referencing the SNU620 IC_50_ values (~20 nM paclitaxel; ~10 nM tepotinib). The ramucirumab concentration used was that at which SNU620 cell viability decreased when ramucirumab was combined with paclitaxel. Cell death was quantitated using the Annexin V-APC/PI apoptosis detection kit (Thermo Fisher Scientific) and CytoFLEX flow cytometer (Beckman Coulter, Brea, CA, USA). The proportions of intact and apoptotic cells were calculated using CytExpert 2.0 software (Beckman Coulter).

### 4.7. Migration Assay

AGS, Hs746T, MKN45, SNU620, and SNU638 cells were seeded into six-well plates at 5 × 10^4^/mL. When confluence was attained, a p-200 pipette tip was used to scrape straight lines through the monolayers. The cells were then washed with phosphate-buffered saline (PBS) and further cultured with or without ramucirumab (10 nM), paclitaxel (20 nM), or tepotinib (10 nM) alone or in combination. After incubation for 2–6 days, the scratch widths were photographed and compared with those on day 0.

### 4.8. Western Blotting

The Western blotting followed a standard procedure. The primary antibodies used were anti-MET (#4560; 1:1000; Cell Signaling Technology [CST], Danvers, MA, USA), anti-phospho-MET (#3077; 1:1000; CST), anti-ECAD (#33195; 1:1000; CST), anti-p-PI3K (#4228; 1:1000; CST), anti-PI3K (#4255; 1:1000; CST), anti-CD44 (#3570; 1:1000; CST), anti-c-MYC (sc40; 1:1000; Santa Cruz Biotechnology, Dallas, TX, USA), anti-p-ERK (#9101; 1:1000; CST), anti-ERK sc514302; 1:1000; Santa Cruz Biotechnology), anti-p-AKT (#4060; 1:1000; CST), anti-AKT (#9272; 1:1000; CST), anti-β-catenin (#610153; 1:1000; BD Biosciences, Franklin Lake, NJ, USA), anti-VEGFR2 (#9698; 1:1000; CST), anti-PD-L1 (#13684; 1:1000; CST), and anti-GAPDH (sc32233; 1:4000; Santa Cruz Biotechnology).

## 5. Conclusions

Compared with ramucirumab-plus-paclitaxel, tepotinib-plus-paclitaxel better inhibited the growth and migration of GC cells with *MET* exon 14 skipping mutations and both *MET* amplifications and phosphorylated MET. Tepotinib-plus-paclitaxel was associated with more cellular apoptosis than ramucirumab-plus-paclitaxel. This in vitro study thus strongly supports the need for clinical evaluation of tepotinib-plus-paclitaxel; this combination efficiently treats GCs with *MET* exon 14 skipping mutations and GCs lacking *MET* amplification but containing phosphorylated MET.

## Figures and Tables

**Figure 1 ijms-25-01769-f001:**
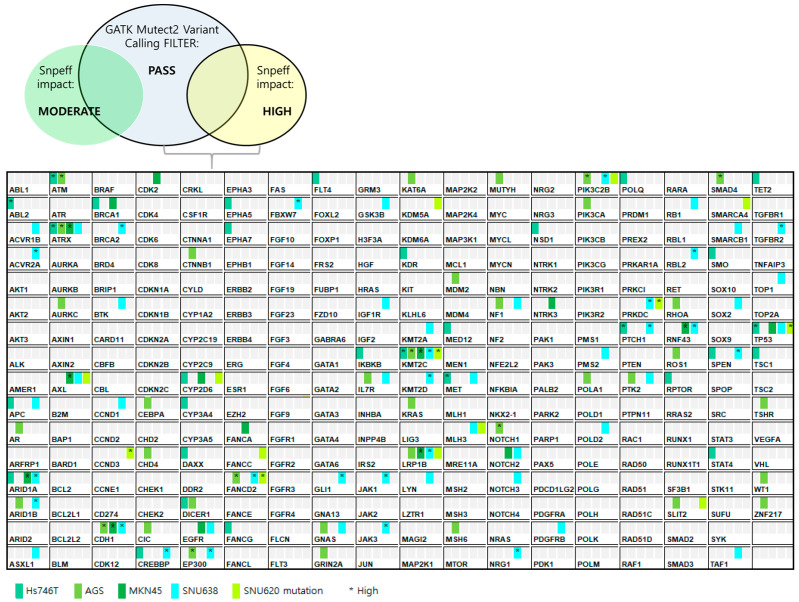
Mutations in 286 genes in five GC cell lines. Mutations were quantitated via targeted next-generation sequencing. All GC-related mutations are shown. Figure 1 was created using Excel 2016 software.

**Figure 2 ijms-25-01769-f002:**
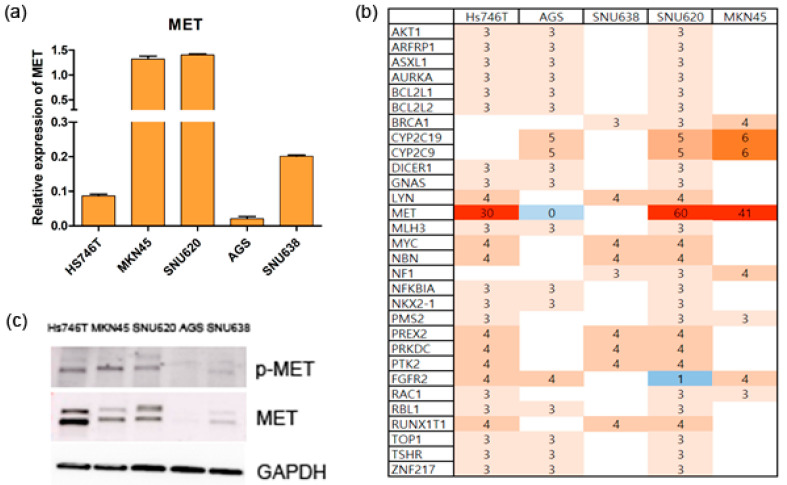
MET gene expression levels and copy numbers, and the MET protein expression levels, in five GC cell lines. (**a**) *MET* mRNA expression levels were determined via quantitative RT-PCR. Quantities of *MET* and *GAPDH* mRNAs were determined in the 5 GC cells using all the tested methods of real-time PCR data analysis. The amount of *MET* mRNA in the 5 GC cells was then divided by the normalization factor (geometric mean of *GAPDH* amounts) of the sample. The values are arithmetical means ± S.E.M., *n* = 3. (**b**) *MET* copy numbers were assessed via targeted NGS. (**c**) The phospho-MET and total MET protein levels were determined via Western blotting.

**Figure 3 ijms-25-01769-f003:**
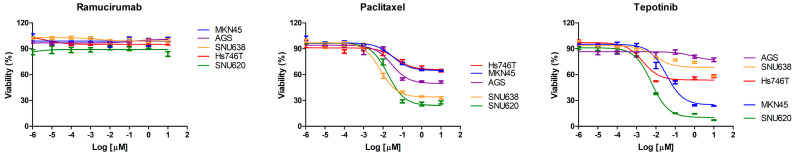
The effects of ramucirumab, paclitaxel, and tepotinib alone on GC viability. Five GC cell lines were treated with various concentrations of ramucirumab, paclitaxel, or tepotinib for 48 h.

**Figure 4 ijms-25-01769-f004:**
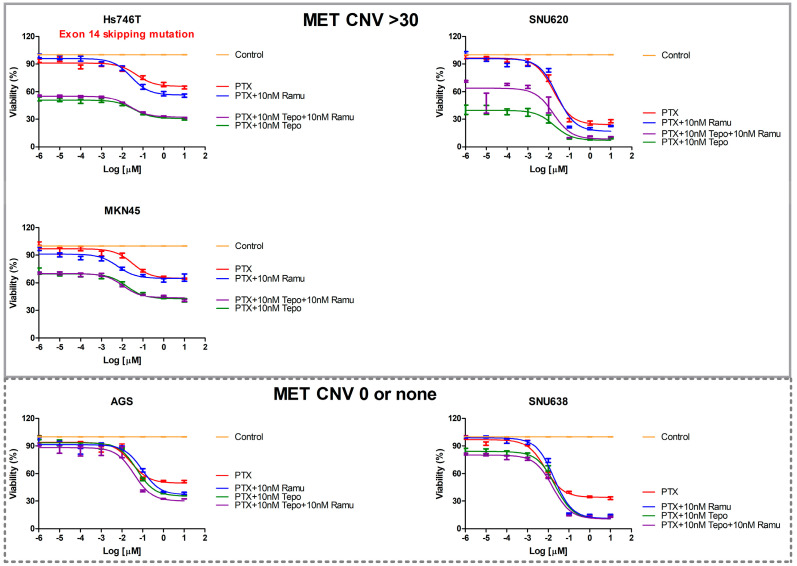
The effects of drug combinations on GC cell viability. GC cells were treated with various concentrations of paclitaxel with or without 10 nM ramucirumab or 10 nM tepotinib for 48 h. Tepo, tepotinib; Ramu, ramucirumab, PTX, paclitaxel.

**Figure 5 ijms-25-01769-f005:**
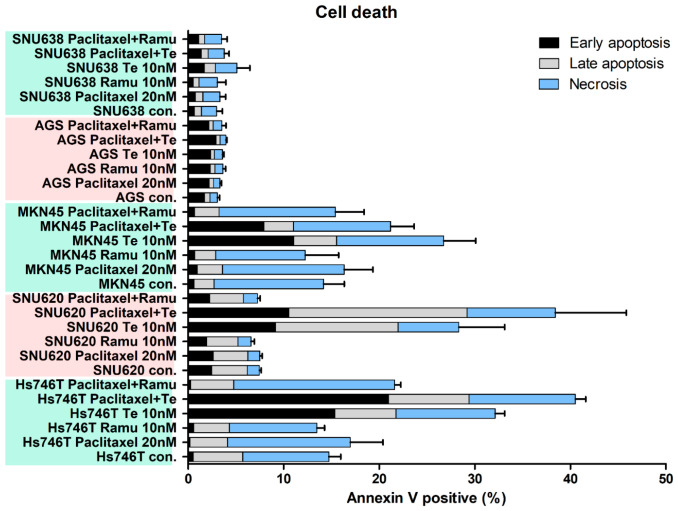
The effects of drug combinations on apoptosis and necrosis in GC cell lines. Hs746T, SNU620, AGS, SNU638, and MKN45 cell lines were treated with 20 nM paclitaxel, 20 nM ramucirumab, or 10 nM tepotinib for 48 h. Con, control; Te, tepotinib; Ramu, ramucirumab.

**Figure 6 ijms-25-01769-f006:**
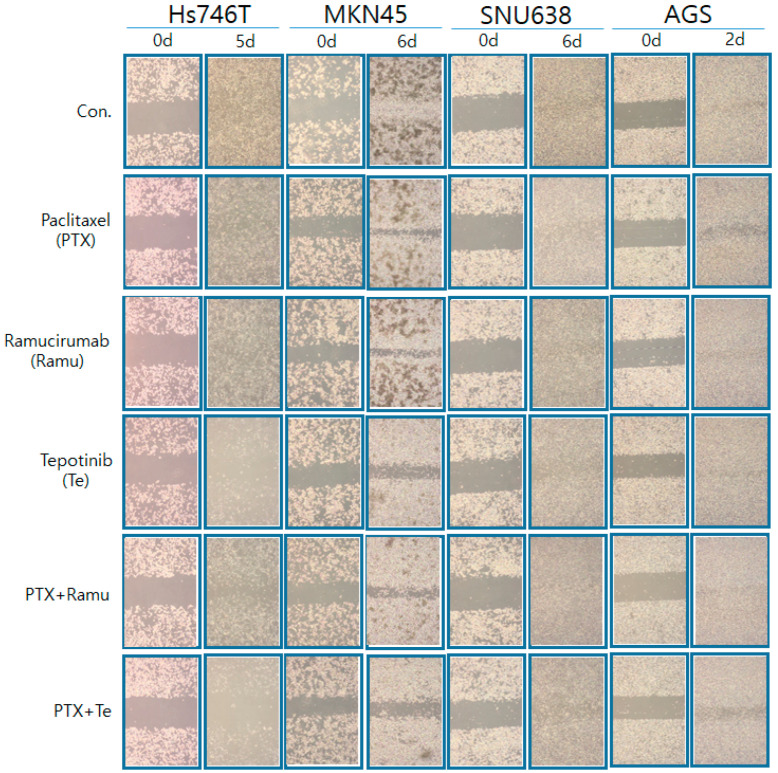
Effects of drug combinations on GC cell migration. A wound-healing assay was used to assess the effects of drugs on the migration of Hs746T, MKN45, SNU638, and AGS cell lines treated with 20 nM paclitaxel, 20 nM ramucirumab, or 10 nM tepotinib for 2–6 days. Con, control.

**Figure 7 ijms-25-01769-f007:**
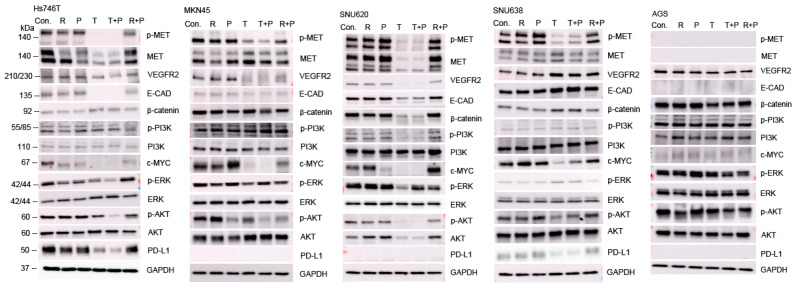
The effects of drug combinations on the expression levels of carcinogenesis-related proteins in GC cells. Con, control; T, tepotinib; Ramu, ramucirumab, P, paclitaxel.

**Table 1 ijms-25-01769-t001:** IC_50_ values of ramucirumab, paclitaxel, and tepotinib in five GC cell lines.

Drug	IC_50_ (nM)				
	Hs746T	SNU620	AGS	SNU638	MKN45
Ramucirumab	-	-	-	-	-
Paclitaxel	-	18.02	28.16	7.272	-
Tepotinib	2.083	5.898	-	-	34.67

## Data Availability

All the data are presented in the body of the manuscript. The data are available from the corresponding author upon reasonable request.

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
