# Peer review of "Comparison of Tepotinib, Paclitaxel, or Ramucirumab Efficacy According to the Copy Number or Phosphorylation Status of the MET Gene: Doublet Treatment versus Single Agent Treatment"

_ijms, 2024, doi:10.3390/ijms25031769_

Round 1
Reviewer 1 Report
Comments and Suggestions for Authors
This study by Sohn et al investigated how copy number variations of the MET gene, MET mutations, and MET gene and protein expression levels in human GC cells affect the susceptibility of such cells to single-agent or doublet treatment. Their results revealed that tep+pac showed better efficacy than ram+pac in inhibiting the growth of GC cells with MET exon skipping mutations and those lacking MET amplification but containing phosphorylated MET. Tep+pac and ram+pac inhibited the growth of GC cells lacking MET amplification or MET phosphorylation in the same level without cell death. However, tep alone or tep+ram was more effective against c-MET-positive GC cells (> 30 copy number variations) than ram, pac alone or ram+pac. However, authors need to revise some major points, as listed below, to make this study complete.
1. This study lacks in vivo data.
2. Why the data in Fig 6 showed different days of result in wound healing?
3. Authors investigated about the necrosis, apoptosis ratio in Fig 5. However, you didn’t explain what methods you used to obtain this figure in result section. Authors should replenish it.
4. If Authors used flow cytometry to obtain Fig 5, you should replenish the bivariant plots.
Comments on the Quality of English LanguageNone
Reviewer 2 Report
Comments and Suggestions for Authors
This study investigates the effectiveness of different anti-cancer drug combinations in treating advanced gastric cancer (GC), with a focus on reducing the adverse effects commonly associated with conventional chemotherapies. The research evaluates the impact of copy number variations of the MET gene, MET mutations, and levels of MET gene and protein expression on the susceptibility of human GC cells to various treatment regimens. They show that tepotinib-plus-paclitaxel is notably effective in inhibiting the growth of GC cells with MET exon 14-skipping mutations and those lacking MET amplification but containing phosphorylated MET. This inhibition is dose-dependent and leads to cell death. Both tepotinib-plus-paclitaxel and ramucirumab-plus-paclitaxel similarly inhibit the growth of GC cells lacking MET amplification or MET phosphorylation, though without inducing cell death. Additionally, tepotinib alone or in combination with ramucirumab was found to be more effective against c-MET-positive GC cells compared to ramucirumab or paclitaxel alone, or the ramucirumab-plus-paclitaxel combination. These in vitro findings suggest a potential preference for tepotinib-plus-paclitaxel in treating certain types of GC cells. My detailed comments are listed as follows:
1. In Table 1, the inclusion of KMT2C requires further justification. Its role in the investigated GC cell lines (overexpression or depletion) has not been explored, which raises the question of its relevance to the study. Clarification or removal of this gene from the discussion may be necessary.
2. Regarding Figure 2A, it is essential to specify the number of biological replicates performed for the real-time PCR analysis. The absence of error bars for HS746T and SNU638 cell lines suggests a lack of data on variability. The figure legend should clearly indicate the meaning of the bars and the number of experiments conducted.
3. In Figure 2C, the gel image shows a slight but noticeable size variation in p-MET protein across the HS746 to SNU620 cell lines, as well as the MET protein itself. This observation should either be explained or the image should be replaced with a more representative western blot.
4. The dataset for the effects of paclitaxel plus tepotinib on cell viability seems to be absent from Figure 3. I
5. Line 121 would benefit from a clearer indication of which cell lines possess MET exon 14-skipping mutations. Additionally, the main text should explicitly state which cell lines respond to the treatments discussed in the results for Figure 4.
6. Figure 4 lacks a vital control showing cell viability in the absence of drug treatment, which is necessary for a baseline comparison.
7. The interpretation of Figure 5 could be expanded to consider the inherent drug sensitivity of different cell lines. Presently, the data does not discount this variable, which could offer an alternative explanation for the observed results.
8. In Line 152, "tepotinib-plus-ramucirumab" should be corrected to "tepotinib-plus-paclitaxel".
9. All western blots in the manuscript should be annotated with the protein sizes to improve clarity and allow for easier interpretation of the results.
10. The methodology section would benefit from a detailed description of how real-time PCR was performed and analyzed, ensuring reproducibility and clarity for the reader.
Comments on the Quality of English LanguageMinor revisions are required.
Reviewer 3 Report
Comments and Suggestions for Authors
The type of the submitted manuscript has been correctly assigned by the Authors as “Communication”. This is a short note, describing how the three studied APIs, either as a single regiment or in a combination, affect the MET-gene related properties.
Despite its length, this communication is interesting an well described. Therefore, I recommend to accept this paper after several corrections listed below.
Line 44, a reference is needed here
Line 45, is it the only ligand of this receptor?
Line 63, this sentence should start in a new paragraph
Figure 2A, table presenting the values presented in this figure, with uncertainties, must be created
Table 2, actually, those values should be presented in nM not μM, due to the low values
Line 188, I don’t agree that those studies failed. They just proved the lack of usefulness of certain therapy. This should be rewritten.
Round 2
Reviewer 2 Report
Comments and Suggestions for Authors
The manuscript can be taken as a great improvement and I recommend it for future publication.